# Extensive Histopathological Characterization of Inflamed Bowel in the Dextran Sulfate Sodium Mouse Model with Emphasis on Clinically Relevant Biomarkers and Targets for Drug Development

**DOI:** 10.3390/ijms22042028

**Published:** 2021-02-18

**Authors:** Rita Bonfiglio, Filippo Galli, Michela Varani, Manuel Scimeca, Filippo Borri, Sara Fazi, Rosella Cicconi, Maurizio Mattei, Giuseppe Campagna, Tanja Schönberger, Ernest Raymond, Andreas Wunder, Alberto Signore, Elena Bonanno

**Affiliations:** 1Department of Experimental Medicine, University “Tor Vergata”, via Montpellier 1, 00133 Rome, Italy; rita.bonfiglio@uniroma2.it (R.B.); manuel.scimeca@uniroma2.it (M.S.); sarafazi@hotmail.it (S.F.); 2Nuclear Medicine Unit, Department of Medical-Surgical Sciences and of Translational Medicine, Faculty of Medicine and Psychology, “Sapienza” University of Rome, 00161 Rome, Italy; filippo.galli@uniroma1.it (F.G.); varanimichela@gmail.com (M.V.); giuseppe.campagna@uniroma1.it (G.C.); alberto.signore@uniroma1.it (A.S.); 3San Raffaele University, via di Val Cannuta 247, 00166 Rome, Italy; 4Saint Camillus International University of Health Sciences, via di Sant’Alessandro, 8, 00131 Rome, Italy; 5UOC Anatomia Patologica, Department of Oncology, USL Toscana Sud-Est, San Donato Hospital, 52100 Arezzo, Italy; filippo.borri@uslsudest.it; 6Interdepartmental Center for Comparative Medicine, Alternative Techniques and Aquaculture (CIMETA), University of Rome “Tor Vergata”, via Montpellier 1, 00133 Rome, Italy; rosella.cicconi@uniroma2.it (R.C.); mattei@uniroma2.it (M.M.); 7Department of Biology, University of Rome “Tor Vergata”, via della Ricerca Scientifica 1, 00133 Rome, Italy; 8Divison of Target Discovery Research and Target Validation Technologies, Boehringer Ingelheim Pharma GmbH & Co. KG, 88387 Biberach an der Riss, Germany; tanja.schoenberger@boehringer-ingelheim.com; 9Immunology and Respiratory Department, Boehringer Ingelheim Pharma GmbH & Co. KG, Ridgefield, CT 06877, USA; ernest.raymond@boehringer-ingelheim.com; 10Division of Translational Medicine and Clinical Pharmacology, Boehringer Ingelheim Pharma GmbH & Co. KG, 88387 Biberach an der Riss, Germany; andreas.wunder@boehringer-ingelheim.com; 11“Diagnostica Medica” and “Villa dei Platani”, Neuromed Group, 83100 Avellino, Italy

**Keywords:** inflammatory bowel disease, dextran sulfate sodium mouse, preclinical model, histopathology, biomarker discovery

## Abstract

This study aims to develop a reliable and reproducible inflammatory bowel disease (IBD) murine model based on a careful spatial–temporal histological characterization. Secondary aims included extensive preclinical studies focused on the in situ expression of clinically relevant biomarkers and targets involved in IBD. C57BL/6 female mice were used to establish the IBD model. Colitis was induced by the oral administration of 2% Dextran Sulfate Sodium (DSS) for 5 days, followed by 2, 4 or 9 days of water. Histological analysis was performed by sectioning the whole colon into rings of 5 mm each. Immunohistochemical analyses were performed for molecular targets of interest for monitoring disease activity, treatment response and predicting outcome. Data reported here allowed us to develop an original scoring method useful as a tool for the histological assessment of preclinical models of DSS-induced IBD. Immunohistochemical data showed a significant increase in TNF-α, α4β7, VEGFRII, GR-1, CD25, CD3 and IL-12p40 expression in DSS mice if compared to controls. No difference was observed for IL-17, IL-23R, IL-36R or F480. Knowledge of the spatial–temporal pattern distribution of the pathological lesions of a well-characterized disease model lays the foundation for the study of the tissue expression of meaningful predictive biomarkers, thereby improving translational success rates of preclinical studies for a personalized management of IBD patients.

## 1. Introduction

Inflammatory bowel diseases (IBDs) are long-life disorders with no effective treatment and whose incidence has been significantly increasing in developing countries [1]. Many animal models have been developed to study its etiology, clinical heterogeneity, and multifactorial nature with limited success [2]. Unfortunately, there are no mouse models that recapitulate all the features of human IBD, but it is worth highlighting that there is considerable genotypic and phenotypic variation even within humans with a similar diagnosis of either Crohn’s disease (CD) or ulcerative colitis (UC) [3]. This creates ongoing challenges for scientists involved in IBD model development and analysis [2,3]. The identification of specific biomarkers and the possibility to compare data using a reliable animal model of colitis could significantly improve the translation of preclinical studies and the outcome of human IBD trials. Therefore, we need more extensive preclinical testing and more careful interpretation of results in animal studies before starting clinical trials in order to ensure safety, avoid unexpected harmful side-effects and improve translation rates [4]. Histopathological descriptions of the frequency and nature of lesions are very often the endpoints in biomedical research conducted in model organisms [5]. Like for the diagnosis of IBD in humans, also for experimental IBD, histological analysis is still the gold standard for evaluating the extent of intestinal lesions and the protective effect of potential drugs [3]. The issue of a reliable histopathological interpretation of data from animal studies is perhaps among the most common sources of error for the assessment of a reproducible and truly translatable preclinical model [6]. Thus, the main aim of this study was to develop a reliable and reproducible IBD murine model based on a careful spatial–temporal histological characterization. Then, we wanted to use such well-established experimental model for the characterization of clinically relevant biomarkers of interest for the development of radiopharmaceuticals for in vivo imaging studies. Indeed, recent advances in imaging technology continue to improve the ability of imaging techniques to non-invasively monitor disease activity and treatment response in preclinical models and in humans with IBD. Nevertheless, to date, very few imaging modalities are routinely available (i.e., CT, NMR, radiolabeled-WBC and FDG PET/CT). It is, therefore, highly clinically relevant to identify new biomarkers that can be the target for new radiopharmaceuticals. Accordingly, as a secondary aim, we performed an extensive preclinical study focused on the in situ expression of clinically relevant biomarkers and targets involved in IBD to advance the drug development pipeline and successful translation towards an even more personalized medicine for affected patients. 

## 2. Results

### 2.1. Clinical Assessment of Dextran Sulfate Sodium-Induced Colitis

Following a pilot study, the optimal experimental setup to obtain a reliable and reproducible Dextran Sulfate Sodium (DSS)-induced IBD model comprised the administration of DSS at 2% for 5 days followed by normal water. We have also observed that DSS-induced disease development occurs more gradually in female animals when compared to male mice. Therefore, we preferred to use female mice, since they are easier to manage, they better tolerate the treatment and ensure more consistency and reproducibility. As shown in Figure 1A,B, colitis development occurred in a slow and steady fashion: mice started to lose weight starting at around five days after DSS exposure; they reached a peak of body weight loss at 5+4 and, after that, they started to recover body mass. Coherently with the curve describing body mass loss, our results showed the beginning of the increase in the Disease Activity Index (DAI) after 4–5 days from DSS intake, its peak at 5 + 4 and an almost total recovery 9 days after DSS removal (Figure 1C,D). As expected, we also observed the shortening of colonic length in 5 + 2 and 5 + 4 mice and a return similar to control group in 5 + 9 mice: (6.52 ± 0.48 cm); 5 + 2 (5.44 ± 0.29 cm), 5 + 4 (5.45 ± 0.32 cm) and 5 + 9 (6.24 ± 0.33 cm). Such results were consistent between two different studies performed following the same experimental procedure. In both experiments, we recorded a body weight loss trend, compatible with animal survival and characterized by a slow and steady onset of the clinical symptoms. Comparisons between body weight loss % and DAI mean values at 5 + 2, 5 + 4 and 5 + 9 obtained from the two experimental runs confirmed that data were consistent between studies in that no statistical differences in terms of body weight loss percentage were detected. In light of these clinical results, we confirmed our model as reliable and reproducible. Thus, the following results are presented by pooling data from the two experimental runs. 

### 2.2. Histopathological Assessment of DSS-Induced Colitis

#### Mucosal Damage

With the aim to provide a rigorous characterization of the DSS-induced model as a relevant model for translational purposes, we performed a spatial–temporal analysis of the established DSS-induced model taking inspiration from the last guidelines promoted by the European Consensus of Crohn’s and Colitis for the Histopathological Diagnosis of Human IBD [7]. To assess the mucosal damage, we developed a scoring system (see Table 1 and Figure 2) by evaluating three main histological characteristics: lamina propria cellularity, architectural damage and epithelial abnormalities (Figure 2). 

As shown in Figure 3A, we observed a significant increase in architectural damage, epithelial abnormalities and lamina propria cellularity at 5 + 2, 5 + 4 and 5 + 9 time points when compared to the No DSS group (architectural damage: No DSS vs. 5 + 2, *p* = 0.0074; No DSS vs. 5 + 4, *p* < 0.0001; No DSS vs. 5 + 9, *p* < 0.0001; epithelial abnormalities: No DSS vs. 5 + 2, *p* = 0.0002; No DSS vs. 5 + 4, *p* < 0.0001; No DSS vs. 5 + 9, *p* = 0.029; lamina propria cellularity: No DSS vs. 5 + 2, *p* < 0.0001; No DSS vs. 5 + 4, *p* < 0.0001; No DSS vs. 5 + 9, *p* < 0.0001. Additionally, a significant increase in architectural damage was observed in both 5 + 4 and 5 + 9 as compared to 5 + 2 group (5 + 2 vs. 5 + 4, *p* = 0.0001; 5 + 2 vs. 5 + 9, *p* = 0.0081) (Figure 3A). The same significant differences were noted also analyzing the lamina propria cellularity (5 + 2 vs. 5 + 4, *p* = 0.0062; 5 + 2 vs. 5 + 9, *p* = 0.0015). A significant decrease in the occurrence of epithelial abnormalities was instead observed in 5 + 9 group as compared to both 5 + 2 and 5 + 4 ones (5 + 2 vs. 5 + 9, *p* < 0.0001; 5 + 4 vs. 5 + 9, *p* < 0.0001). 

Overall, we found microscopic features highly suggestive for the diagnosis of UC: heavy mucosal and transmucosal inflammatory infiltrate (mainly lymphocytes and plasma cells at the base of shortened mucosal crypts or in the lower layers of lamina propria), but no granulomas and transmural inflammation or prominent lymphoid aggregates (typical of Crohn’s colitis). Additionally, the overall levels of inflammatory infiltrate (including lymphocytes, plasma cells, eosinophils and neutrophils) significantly increased from the basal condition in all DSS time points by reaching the peak at 5 + 2 and significantly decreasing from it at 5 + 4 and at 5 + 9. Thus, according to our score criteria, the significantly higher levels of inflammatory infiltrate recorded at 5 + 2 with respect to 5 + 4 and 5 + 9 time points indicate that a higher presence of neutrophils (score = 3) characterizes the composition of 5 + 2 inflammatory infiltrate. Nevertheless, an evident presence of neutrophils (score = 3) was not frequent: unlike active phase of human UC, murine acute colitis does not show frequent foci of cryptitis or crypt abscesses separated from ulcers, favoring the presence of neutrophils more often in lamina propria or in the context of ulcerative processes. Moreover, in many cases, the acute inflammatory infiltrate could be present both in ulcerated mucosa and in underlying pericolic adipose tissue, without evident effacement of interposed tunica (a feature more compatible with Crohn’s ulcerative mucosal damage, which might be characteristically transmural, sometimes with fistulas formation). The contemporary presence of the highest levels of epithelial damages accompanied by a focally active inflammatory infiltrate clearly indicates that, according to our DSS model, the 5 + 2 time point is to be considered the temporal window of an early active acute disease (where the acute infiltrate, i.e., neutrophilic component, is more evident). In comparison to 5 + 2, at time point 5 + 4, architectural damage increased, whereas epithelial damage persisted accompanied by a less evident acute inflammatory infiltrate. At 5 + 9, we found that, whereas epithelium significantly recovers from the damage, an abnormal tissue architecture persists when compared to 5 + 4 time points. Here, the presence of acute inflammation is significantly lower than at 5 + 2, and atrophy characterizes the dominant pathological feature of lesions, while a substantial reduction in ulcers and erosions is appreciable. Accordingly, a significant presence of chronic inflammatory infiltrate resides in the areas of repairing mucosa. Therefore, unlike human chronic inactive morphology, in DSS mice, features of chronic mucosal damage (e.g., crypt distortion, Paneth cell metaplasia in distal colonic mucosa, heavy inflammatory infiltrate and basal plasmocytosis) were not present alone, but they were more often observed, focally, at margins of overt and clear active disease. Thus, according to histology, the clinical remission, observed in DSS mice at 5 + 9, did not exactly correspond to a quiescent phase of the human disease, since chronic inactive morphology appears, focally, albeit rarely, accompanied by lesions with active inflammatory infiltrate.

### 2.3. Anatomical Distribution of the Mucosal Damage

Concerning the anatomical distribution of the mucosal damage, it was interesting to note that biopsies from the distal part of the colon, at any time, presented significantly higher values of all three examined histological features (Figure 3B–D).

Finally, analysis of colonic mucosa, evaluated in terms of the number of pathological samples per whole colon, at the three different time points, revealed that the involvement of the colonic mucosa progressively increased both in the proximal (from 48.65% at 5 + 2 to 62.13% at 5 + 4) and distal colon (95.75% at 5 + 2 to 100.00% at 5 + 4), reaching the complete involvement of the distal colonic mucosa at the 5 + 4 time point. Notably, the anatomical distribution of the lesions (in terms of number of pathological colon rings/whole colon) effectively explains why during this temporal window mice suffered the maximum peak of weight loss, thereby showing the worst DAI at this time point (see Figure 1). Similarly, the incidence of pathological rings on the whole colon (in particular, 58.63% of pathological rings/proximal colon; 66.67% of pathological rings/ distal colon) could explain the apparent clinical remission showed by 5 + 9 mice. Indeed, although the morphological appearance of the lesion does not exclude the presence of areas of focally active disease (albeit in the context of a dominant chronic inactive morphology), the recovery of a large amount of colonic mucosa could explain the clinical remission measured in terms of body weight mass and DAI (Figure 1).

### 2.4. Immunohistochemistry

Immunohistochemical analysis was performed to assess the tissue expression of inflammatory markers of clinical interest for IBD in a well-established murine model along different time points. The tissue expression levels of TNF-α, α_4_β_7_, VEGFRII, GR-1, CD25, CD3, IL-12p40, IL-23R, IL-17A, IL-36R and F4/80 were evaluated by assigning a score from 0 to 3 according to the number of positive cells. Analysis of TNF-α showed a significant group effect (*p* = 0.0002) with a significant increase in the number of TNF-α positive cells in colonic mucosa from DSS mice with respect to no DSS. In detail, our post-hoc analysis (Tukey test) showed the following significant differences: (a) No DSS vs. 5 + 2, *p* = 0.0007; (b) No DSS vs. 5 + 4, *p* = 0.0004; (c) No DSS vs. 5 + 9, *p* = 0.0012 (Figure 4A–E). Additionally, for α_4_β_7_ expression levels, we detected a significant group effect (*p* = 0.033) when DSS mice were compared to control mice. A Tukey test revealed a significant difference in the number of α4β7 positive cells between No DSS and 5 + 9 (*p* = 0.025) (Figure 4F–J). Immunohistochemical analysis of VEGFRII showed a significant group effect (*p* = 0.0006) and also a significant increase in the number of VEGFRII positive cells in all experimental groups compared to No DSS mice (No DSS vs. 5 + 2, *p* = 0.0025; No DSS vs. 5 + 4: *p* = 0.039; No DSS vs. 5 + 9, *p* = 0.0006) (Figure 4K–O). A significant difference in the number of neutrophils (Gr-1 positive cells) among the experimental groups was found (group effect *p* < 0.0001). In detail, we observed the following differences: No DSS vs. 5 + 2, *p* < 0.0001; No DSS vs. 5 + 4, *p* = 0.0041; No DSS vs. 5 + 9, *p* = 0.0084 (Figure 5A–E). Of relevance, Gr-1 significantly increased from the basal condition in all DSS time points by reaching the peak at 5 + 2 and significantly decreasing from it at 5 + 4 and at 5 + 9 (5 + 2 vs. 5 + 4, *p* = 0.0058; 5 + 4 vs. 5 + 9, *p* = 0.0028). Concerning the analysis of CD25 positive cells, a significant group effect was found (*p* = 0.011). A Tukey test showed a significant increase in the number of CD25 positive cells in 5 + 2 with respect to No DSS (*p* = 0.0015) and in the 5 + 9 group if compared with No DSS (*p* = 0.018) (Figure 5F–J). Considering CD3 tissue expression levels, a significant group effect was observed among the experimental groups (*p* = 0.028). However, post hoc analysis showed a significant increase in the number of CD3 positive cells only in 5 + 9 if compared with the No DSS group (*p* = 0.023) (Figure 5K–O). Additionally, for IL-12p40, we found a significant group effect (*p* = 0.03), and in detail, there was an increase in 5 + 4 and 5 + 9 groups compared to No DSS mice (No DSS vs. 5 + 4: *p* = 0.044 and No DSS vs. 5 + 9: *p* = 0.041), as shown in Figure 6A–E. Lastly, from the group effect and post hoc analyses performed on the expression levels of IL-23R, IL-17A, F480 and IL-36R, no significant differences were observed (Figure 6F–I). 

## 3. Discussion

As the pathogenesis of IBD is still incompletely understood, therapeutic approaches have mostly been limited to the rather unspecific suppression of the adaptive immune system. Recent advances in understanding the underlying immune-pathogenetic mechanisms of IBD have led to the development of biological therapies, which aim to selectively inhibit crucial mediators of the inflammatory process [8,9]. However, in designing these novel clinical approaches, it should be kept in mind that IBDs are heterogeneous, in both clinical manifestations and response to therapy, and therefore, it is highly likely that no drug will work in all patients [10]. Predictive biomarkers are used to identify individuals within the larger IBD patient population who are likely to have a positive response to treatment with a specific therapeutic drug [3]. Therefore, a key unmet need is the establishment of predictive biomarkers that can make the difference between a successful or failed trial. A common thread in most of these research efforts is the need for appropriate animal models, or rather, the need for animal models that share pathogenic features and/or utilize molecular pathways of interest to enable hypothesis testing or even to conduct preliminary hypothesis generating studies. Ideally, a disease model should closely parallel the human disease in clinical manifestations, pathophysiology and response to existing therapeutic reagents. However, this is rarely possible with complex human diseases, such as IBD; therefore, research efforts aim to identify the specific features we need to model rather than trying to recapitulate the entirety of human disease in an experimental animal. Chemically induced models of colitis, such as DSS, induce acute damage to the epithelium. Probably, upon loss of barrier integrity, bacteria rapidly penetrate the epithelium and mediate both direct and inflammation-associated toxicity [11]. From this standpoint, DSS colitis represents a powerful tool for assessing gut integrity, host responses to microflora mediated by intestinal epithelial cells and innate immune cells. Since histopathology is still the gold standard for the diagnosis and management of IBD patients, we chose to characterize the histopathological features of a model of acute inflammation elicited by administering 2% DSS to female C57BL/6 mice of 8–10 weeks for 5 days, followed by 2, 4 or 9 days of normal water. In our experience, this experimental setup ensured the necessary characteristic of reproducibility of the model to consider it a reliable tool for specific preclinical purposes, such as predictive biomarker development. The description of the clinicopathological features of the DSS-induced model is not new [12,13]. Nevertheless, descriptions of histopathological methods to evaluate colitis in mouse models often do not include enough details to allow readers to understand the methodology and do not reflect a pathologist’s assessment. The histopathological interpretation of experimental animals is an ongoing concern and perhaps among the most common sources of translational failures. Thus, in this work, we wanted to develop a reliable and reproducible IBD murine model based on a careful spatial–temporal histological characterization by taking inspiration from the last histopathological guidelines for human IBD. In agreement with the European Crohn’s and Colitis Organisation (ECCO) guidelines, we performed multiple colon biopsies (in our case, we sampled the entire colon), keeping track of the anatomical site of origin of each biopsy (thanks to the employment of a handicraft tissue array) and by examining multiple sections (*n* = 3) for each biopsy “since lesions may be focal”. The sampling of the whole colon, cut into rings, was preferred to as the Swiss roll technique [14], because it permits a better assessment of architectural abnormalities, as it occurs for the diagnosis of human disease for which colon specimens are oriented “submucosal downside”. All tissue samples were immediately fixed, as recommended for diagnostic routine, with an original squeezing technique (see methods), preferable to the most commonly used flushing of colons with PBS, because immediate fixation optimally preserves the gross and microscopic anatomy of the tissue. Histological score assessment was performed, accounting for three main categories sufficient to reflect the severity of histopathology independent of the localization and overall extent of inflammation: (1) quality and dimension of inflammatory cell infiltrates, (2) overall mucosal architecture and (3) epithelial changes [15]. Scoring schemata were defined along the specified criteria for each of the three categories (Table 1). Overall, the active phase of DSS-induced colitis was represented by mucosal atrophy with ulcers, erosions and granulation tissue, with or without the evident presence of neutrophilic acute inflammation. Like in human colitis, the presence of neutrophils also indicates disease activity. Nevertheless, as reported elsewhere, in the murine model, the presence of neutrophils was not always evident. Unlike the active phase of human ulcerative colitis, murine acute colitis did not show frequent foci of cryptitis or crypt abscesses separated from ulcers, favoring the presence of neutrophils more often in lamina propria or in the context of ulcerative process [16]. To avoid over or under-diagnoses of active disease, we considered ulcers, erosions and mucosal atrophy as contributors to histological criteria for the active phase of DSS-colitis, because in human disease, neutrophils alone are not considered specific in IBD diagnosis without architectural modifications, especially in UC [17]. Nevertheless, we recommend the close inspection of tissue sections showing granulation tissue and surface erosions without the clear presence of neutrophils, as acute inflammatory infiltrate may be patchy. Moreover, colonic mucosal orientation is of crucial importance in evaluating epithelial damage and, more importantly, architectural features; differentiating epithelial ulcer from erosion can be challenging, since epithelial erosion is more superficial than ulcer, which deeply involves intestinal mucosa. Glandular shortening and dropout might be misinterpreted in the absence of reliable intestinal wall orientation in histological sections. Therefore, we recommend considering all histological features together and the careful examination of intestinal samples in the case of remarkably discordant features. Unlike human ulcerative colitis, neutrophils could be present both in ulcerated mucosa and in underlying pericolic adipose tissue, without evident effacement of interposed tunica muscularis (a characteristic feature of Crohn’s ulcerative transmural damage). This could be explained by the fact that the mouse colonic wall is thinner than that of a human, and transmural migration of neutrophils could occur more easily, together with characteristic acute and intense mucosal damage typical of DSS-induced colitis. Even though a complete transposition of human pathological pattern into murine mucosa was neither obvious nor correct, because of species-specific differences, altogether, clinical reports and histological data led us to formulate, in line with other literary data [18], a diagnosis compatible with ulcerative colitis. Of note, the application of the new scoring method for the assessment of the mucosal damage and the evaluation of the anatomical distribution of the lesions along the whole colon allowed us to discriminate not only the IBD type (UC vs. CD) but also the different phases of the disease over time. Indeed, we could refer to the three time points analyzed here as: (a) an early phase (5 + 2), characterized by an acute active disease mainly involving the rectum, and speeding proximally with gradually decreasing severity of the damage without skipping areas; (b) an advanced phase (5 + 4), characterized by an acute active disease extended through almost the entire colon; (c) a recovery phase (5 + 9), characterized by a focally active disease dispersed into a prevalent healthy/chronic inactive mucosa. The application of a careful scoring method for the assessment of the mucosal damage and the evaluation of the anatomical distribution of the lesions along the entire colon could offer a more reliable translational tool for the interpretation of preclinical data. Additionally, the development of well-designed experimental protocols could benefit from rigorous knowledge of the experimentally induced murine pathology, from analogies and differences with its human counterpart. For the first time, in this paper, a well-established DSS-induced IBD model was used for the study of the temporal in situ expression profile of biomarkers; some of these are already well known and involved in the pathogenesis or therapy of human IBD, which are also highly clinically relevant to identify new biomarkers against which we can produce radiopharmaceuticals: TNF-α, α_4_β_7_, VEGFRII, GR-1, CD25, CD3, IL-12p40, IL-23R, IL-17A, IL-36R and F480. TNF-α is chronically elevated, locally and systemically, in patients with IBD [19,20]. In line with this evidence, in our CU-like experimental model, the tissue expression of TNF-α was significantly higher in DSS mice with respect to control mice, and its expression levels were similarly maintained along the time points analyzed. Beyond TNF-α blockers, numerous other compounds are in the pipeline or have been recently approved. Another crucial pathway in the inflammatory cascade is the aberrant migration of cells from the systemic circulation into the gastrointestinal tract [21]. Therefore, different anti-adhesion molecule therapies have been developed to expand the therapeutic possibilities in IBD, among which an antibody against the α_4_β_7_ integrin (vedolizumab) was the first to be approved for IBD treatment [22]. The α_4_β_7_ integrin and its ligand, the MadCAM-1, have been of great interest, since they are found exclusively on the gut-homing lymphocyte subpopulations and in the intestinal mucosa, respectively [23]. The aberrant migration of cells from the systemic circulation into the gastrointestinal tract found in human IBD is also detectable in our murine model, measured by the significant increase in positive α_4_β_7_ cells in DSS mice at 5 + 9, as compared to the control group. The significantly higher levels of positive cells at 5 + 9 with respect to No DSS mice suggest a later increase in α_4_β_7_ tissue expression in a model of acute inflammation like our DSS-induced colitis, concomitant to the recovery phase. Angiogenesis is another important process related to the pathogenesis of IBD, and it is a biological phenomenon coordinated by several proangiogenic molecules and their receptors, such as those of the VEGF family [24,25]. Among them, the expression of VEGFRII seems to increase in intestinal mucosal samples of IBD both in human patients and in mice with experimental colitis [26]. Our results seem to confirm these data. A significant increase in the number of VEGFRII positive cells was found in DSS mice when compared to No DSS ones, although we did not observe differences along the analyzed time points. As already studied for human cancers [27], this further supports the research of new targeting molecules capable to detect the in vivo expression of VEGFRII in the bowel. 

In experimental animal models of IBD, controversial data on the role of neutrophils have been reported. Some studies showed that neutrophil depletion by anti-neutrophil antibodies ameliorates colitis induced by DSS [28,29], whereas others indicated a beneficial role of neutrophils during colitis, with exacerbation of inflammation after their depletion [17,30]. This discrepancy is probably due to the use of different rodent models of colitis.

In our experimental conditions, Gr-1 expression supports the evidence that neutrophils play a role in the activity of the disease. More specifically, in line with our study on the inflammatory infiltrate (evaluated by HE analysis), Gr-1 expression levels confirmed that neutrophils participate in all phases of the induced disease, thereby conditioning its activity. Notably, the earliest phases of the DSS-induced disease (5 + 2) are those characterized by the highest levels of the neutrophilic component (Gr-1 positive cells) in the context of the inflammatory infiltrate. This information could be of great importance when planning the development of new drugs targeting these biomarkers. For example, many neutrophil-specific Positron emission tomography (PET)/ Single-photon emission computed tomography (SPECT) imaging agents have been developed for application in human IBD [31,32], and our data provide a strong rationale for their use. They could also contribute to identifying the most appropriate timing for neutrophil detection and targeting to develop new drugs and evaluate their efficacy in a non-invasive manner. Similarly, several PET agents have been proposed to monitor immunotherapy both in IBD [33] and in cancer [34,35] through in vivo imaging of CD25. Interestingly, an increase in the number of CD25 positive cells was found at 5 + 2 and at 5 + 9 when compared to the No DSS group. Finally, an increase in CD3 positive cells was observed in the 5 + 9 group, suggesting that longer times are required to establish a chronic inflammatory response. The discovery that IL-12 and IL-23, 2 heterodimeric cytokines sharing the common p40 subunit, are over-produced in the inflamed intestine of IBD patients, together with data emerging from studies in murine models of colitis, suggest that blockade of the p40 subunit may be therapeutic in IBD [36]. In line with literary data, we found a statistically significant difference in IL12p40 expression between control mice and DSS mice (5 + 4 and 5 + 9 groups). The available data in humans support the pathogenic role of IL-12/IL-23 in the gut, although much work remains to be conducted in order to delineate the optimal scenario in which IL-12/IL-23/Stat inhibitors should be used in IBD and ascertain whether IL-12p40 blockers could help manage some subsets of IBD patients. Monitoring recurrence and evaluating response to therapy are important aspects of clinical decision making in the treatment of IBD. Relapses are often difficult to predict. The goal of disease monitoring is to identify patients at risk for relapse in order to treat earlier, with the hope of maintaining remission and avoiding irreversible bowel damage. Imaging modalities can help in therapy decision making, management and follow-up through the evaluation of the expression of specific molecules, leading to the development of personalized therapies. With the advent of personalized medicine and the increase in the targeted drugs or imaging agents, providing new biomarkers and a reliable animal model are both key factors for a better management of human diseases. Our DSS model could be a useful resource for researchers that aim to develop and test new drugs or imaging agents that target TNF-α, α_4_β_7_ or any other analyzed markers. In this way, it is possible to refine experiments and reduce the number of animals but maintain the optimal reproducibility and consistency of data that could be easily translated into human trials.

## 4. Materials and Methods

### 4.1. Ethical Statement 

Experimental procedures were previously approved by the OPBA (the institutional animal-welfare body) of the University of Rome “Tor Vergata” and were then authorized by the Ministry of Health with identification code 188/2016-PR of 22 February 2016, the study was carried out in accordance with the Italian and European regulations on the protection of animals used for scientific purposes (D.L.vo 26/2014; C.E. 63/2010).

### 4.2. Mice and Maintenance

C57BL/6 female mice (8 to 10 weeks of age) were purchased from Harlan Laboratories, srl (Lesmo, MB, Italy). Animals were housed in the animal care facility (CIMETA) of the University of Rome “Tor Vergata” Roma, Italy in collective cages at 20 ± 2 °C under a 12 h light/dark cycle and with 4RF25 GLP food (Mucedola s.r.l., Settimo Milanese, MI, Italy) and water provided ad libitum. They were allowed to acclimate to these conditions for at least 7 days before performing any experiments. 4.3. Induction and Evaluation of DSS Colitis.

To establish the optimal experimental conditions to develop a reliable and reproducible preclinical model, we performed a pilot study by exploring the protocol of DSS-induced colitis several times in different set-ups (i.e., by comparing the administration of 3% DSS in male vs. female C57BL/6 mice and by comparing the administration of 2% DSS vs. 3% DSS in female mice; data not shown). In this study, colitis was induced by adding 2% DSS (35–50,000 kDa; MP Biomedicals, Illkirch, France) to drinking water for 5 days followed by: (a) 2 days of recovery with normal drinking water (*n* = 5), (b) 4 days of recovery with normal drinking water (*n* = 5), (c) 9 days of recovery with normal drinking water (*n* = 5). Control group (NO DSS; *n* = 5 mice/group) received normal drinking water ad libitum. To assess reproducibility, this experimental protocol was performed twice. All animals were matched for age, equilibrated for body weight and subdivided in groups of 5 mice per cage. On the day of DSS administration (day 0), each mouse from the control and experimental groups was weighed and, if required, average group weight equilibrated to eliminate any significant difference between groups. In accordance with Institutional Animal Care and Use Committees (IACUC) recommendations, loss of 25–30% of initial body weight was considered as an endpoint criterium followed by euthanasia, also in accordance with institutional guidelines. All animals were monitored daily, and the clinical score (CS) was assessed by the following parameters: individual behavior (normal, ruffled fur or altered gait, lethargic or moribund) and body weight, stool consistency and water consumption per cage. The disease activity index (DAI) was determined based on body weight loss (one point for each 5% loss of weight), stool consistency/cage (0, normal; 2, formed but very soft; 4, liquid) and presence of blood in the stools (0, none; 1, if present on the cage of around the anus) [37]. The percentage of body weight loss was calculated in relation to the starting weight using the formula: [(Weight on day X − Initial weight)/Initial weight] × 100 [38]. Animals were sacrificed by cervical dislocation in correspondence to the three time points: 5 days DSS 2% plus 2, 4 or 9 days with normal water (named as 5 + 2, 5 + 4, 5 + 9, respectively). 

### 4.3. Tissue Processing

After the sacrifice by cervical dislocation, each colon was excised, examined for length (measured from the anus to the top of the cecum) and immediately fixed in a 10% (w/v) formalin solution for further analyses. After 24 h, fixed colons (from the cecum to the anus) were divided into four equal parts. Once the subdivision had been carried out, the four portions of the colon were immerged in water, held by thumb and index at one end to give the organ light pulsating pressures able to free the intestinal lumen from the feces (squeezing), thus simulating the physiological movement of intestinal peristalsis. This squeezing technique was preferred to the flushing of colons with phosphate buffered saline because immediate fixation followed by squeezing optimally preserved the gross and microscopic anatomy of the tissue. Each of the four portions of the intestine, duly cleaned from the feces, was finally divided into rings of 5 mm. All procedures adopted along this pre-analytical phase were performed paying attention to maintain both lumen exposure and the information on the anatomical site of origin of each colon biopsy.

### 4.4. Tissue Staining

For histological analysis, formalin fixed paraffin embedded colonic tissues were sectioned (4 µm thickness), mounted on glass slides and deparaffinized. Slices were stained by using standard hematoxylin and eosin (H&E) [39]. 

### 4.5. Scoring System

To provide a rigorous characterization of the DSS-induced model, as a relevant model for the translation of mice to human disease, we performed a spatial–temporal analysis inspired by the latest guidelines released by the European Consensus of Crohn’s and Colitis for the Histopathological Diagnosis of Human IBD [7]. Accordingly, the entire colon from each mouse was grossed and evaluated ring by ring at three levels. We analyzed all mouse colons by light microscopy paying attention to two specific and fundamental pathological aspects: the anatomic distribution of the lesions (proximal/distal colon) and the definition of the main microscopic features of the intestinal mucosal damage: lamina propria cellularity, architectural damage and epithelial abnormalities (Table 1).

All specimens from proximal (from cecum: colon rings from the first two portions) and distal parts (from cecum: colon rings from the last two portions) of the entire mouse colon were stained with H&E and evaluated on three levels. Scores from 0 to 3 were attributed to each of the three histological features mentioned above according to the increasing severity of the damage. Readings were performed in double-blind by two expert pathologists. 

### 4.6. Immunohistochemistry 

We employed immunohistochemical techniques to characterize the in situ expression of clinically relevant biomarkers described for IBD in our experimental model (see Table 2). Briefly, antigen retrieval was performed by the pressure cooker method (2100 Retriever; Aptum Biologics Ltd., Southampton, United Kingdom) on 3 μm-thick paraffin sections using EDTA citrate pH 7.8 or citrate pH 6.0 buffers (20 min at 120 °C). Sections were then incubated for 1 h at room temperature with primary antibodies (listed in Table 2). Washing steps were performed with PBS/Tween20 pH 7.6. Reactions were revealed by an HRP - DAB Detection Kit (UCS Diagnostic, Rome, RM, Italy) and counterstained with hematoxylin. To perform immunohistochemical analysis, we digitalized all slides (Iscan Coreo, Ventana, Tucson, AZ, USA). Images were captured by using ImageView Software. Immunohistochemical positivity was evaluated on digital images by two blind observers who assigned a score (0–3) by counting the number of positive cells in 4 fields at 20× (score 0: positive cells ≤ 2; score 1: 3 < positive cells ≤ 30; score 2 (31 < positive cells ≤ 100), score 3 (positive cells > 100)). Considering the intensity and the number of positive cells detected following the immunoreactions for TNF-α e CD3, the score (0–3) was attributed by establishing different cut-offs (score 0: positive cells ≤ 2; score 1: 3 < positive cells ≤ 50; score 2 (51 < positive cells ≤ 200), score 3 (positive cells > 200)). To assess the background of immuno-staining, for each reaction, we included a negative control by incubating the sections with secondary antibodies (HRP) and a detection system (DAB). According to the manufacturer’s instructions, reactions were set up by using specific control tissues. 

### 4.7. Statistical Analysis

Comparisons among groups (No DSS, 5 + 2, 5 + 4, 5 + 9) and the variables (IL-17, TNF-α, VEGFRII, IL-23R, Gr-1, F480, α4β7, CD25, CD3, IL-12p40 and IL-36R) were tested by the General Liner Model (GLM) method. A Shapiro–Wilk test was used to verify the normality of distribution of residuals/variables, whereas the homoscedasticity was verified by Levene and Brown–Forsythe tests. Post-hoc analysis was performed by a Tukey test. Comparisons among groups (No DSS, 5 + 2, 5 + 4, 5 + 9) and the variables (Architectural damage, Epithelial Abnormalities and Lamina Propria Cellularity) were evaluated by a Kruskal–Wallis test, and post-hoc analysis was verified by a Steel–Dwass test.

In the “5 + 2”, “5 + 4” and “5 + 9” experiments, comparisons of “Proximal” vs. “Distal” relative to architectural damage, epithelial abnormalities and lamina propria cellularity were tested by a Mann–Whitney test. A *p* ≤ 0.05 was considered statistically significant. Statistical analysis was performed using SAS v. 9.4 and JMP v.15 (SAS Institute Inc., Cary, NC, USA).

## 5. Conclusions

Across all trials with approved biological agents in IBD, about 40% of the patients do not respond to treatment with the applied biological substance, and subsequent repeated treatments produce a therapeutic effect in only a small group of these patients. IBD is a highly complex and chronic disease that requires a personalized therapeutic approach as a standard of care for the management of patients. Knowing the spatial–temporal pattern distribution of the pathological lesions of a well-characterized disease model lays the foundation for the study of the tissue expression of meaningful predictive biomarkers that may allow us to individualize specific therapies based on molecular level analysis, thereby improving translational success rates of preclinical studies for the personalized management of IBD patients.

## Figures and Tables

**Figure 1 ijms-22-02028-f001:**
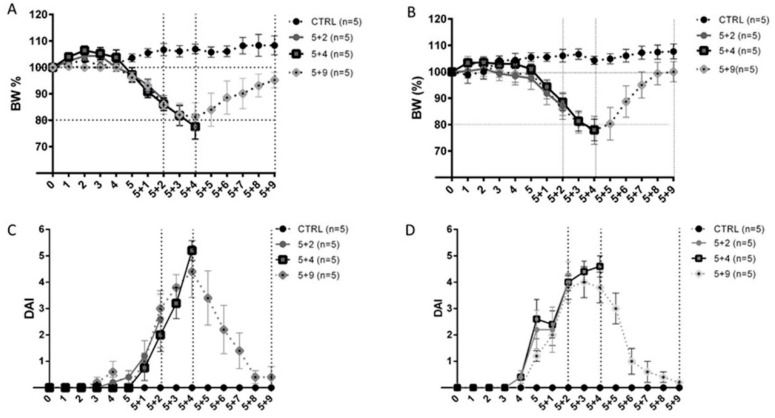
Clinical evaluation of Dextran Sulfate Sodium (DSS)-induced colitis. (**A**,**B**) Graphs describe body weight loss percentage (BW%) of mice daily registered Figure 5 experimental group) used in two distinct experimental runs. (**C**,**D**): Disease Activity Index (DAI) score calculated daily from mice (*n* = 5/experimental group) used in two distinct experimental runs.

**Figure 2 ijms-22-02028-f002:**
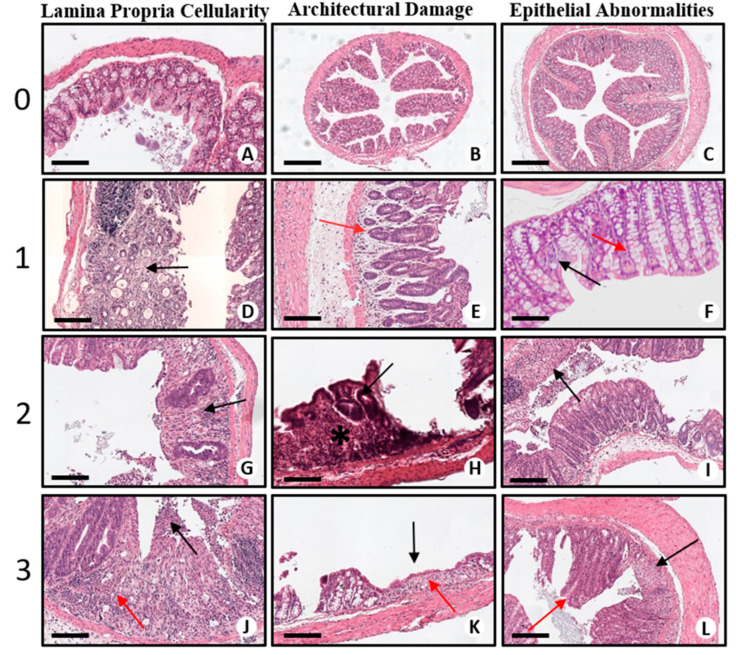
Histological scoring system. (**A**) Normal colonic mucosa with barely visible lamina propria and rare inflammatory cells (lymphocytes and occasional plasma cells). Scale bar represents 200 µm. (**B**) Normal colonic mucosa with linear crypts and regular glandular arrangement. Note that there is not any space between bottom of colonic crypts and muscularis mucosae. Scale bar represents 500 µm. (**C**) Normal colonic mucosa without any sign of epithelial damage: glands show regular profile, and their mucinous compartment is preserved. Scale bar represents 500 µm. (**D**) Colonic mucosa with mild increase in chronic inflammatory infiltrate (black arrow), mainly characterized by lymphocytes and plasma cells. Scale bar represents 200 µm. (**E**) Colonic mucosa with mild glandular distortion: to note the irregular glandular profile and characteristic doubled bottom (red arrow. Scale bar represents 200 µm. (**F**) Colonic mucosa with reactive epithelial alterations: mucin depletion (black arrow) and repairing/reactive changes with glandular hyperplasia and nuclear hyperchromasia (red arrow). Scale bar represents 200 µm. (**G**) Colonic mucosa with striking increased lamina propria cellularity: dense inflammatory infiltrate (black arrow) characterizes this part of colonic mucosa, along with glandular rarefaction. Scale bar represents 200 µm. (**H**) Evident glandular profile distortion (arrow) with glandular rarefaction and crypt shortening (asterisk), involving ≤50% of mucosal thickness. Scale bar represents 200 µm. (**I**) Colonic mucosa with superficial epithelial erosions (arrow) with superficial fibrin exudate and reactive epithelial changes. Scale bar represents 200 µm. (**J**) Striking presence of acute and chronic inflammatory infiltrate (red arrow); there is neutrophilic infiltration mixed with fibrin and necrotic debris right above ulcerated mucosa (arrow). Scale bar represents 200 µm. (**K**) Total epithelial atrophy: there is absence of glands and mucosal flattening (arrow), glands in lamina propria are replaced by collagen and scant inflammatory cells (red arrow). Scale bar represents 200 µm. (**L**) Colonic mucosa with ulcers (epithelial necrosis involving the total thickness of the mucosa (arrow): note the abrupt total loss of glands replaced by granulation tissue and inflammatory infiltrate; the remaining mucosa shows polypoid reactive–regenerative changes (red arrow). Scale bar represents 200 µm.

**Figure 3 ijms-22-02028-f003:**
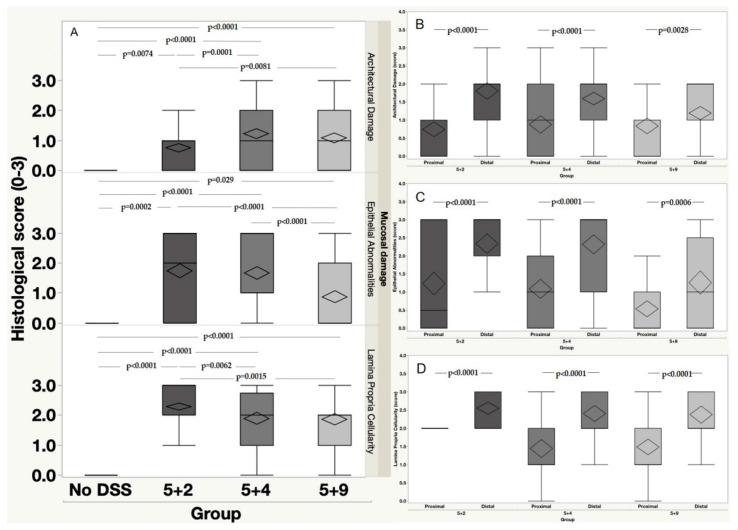
Anatomical distribution of mucosal damage. (**A**) Graph describes the temporal profile of the mucosal damage evaluated at three different time points (5 + 2, 5 + 4,5 + 9) compared to control group (No DSS; *n* = 10 mice/group) according to three morphological categories (lamina propria cellularity, epithelial abnormalities and architectural damage) scored on the basis of histological scores described in Table 1. (**B**–**D**) Graph describes the temporal profile (5 + 2, 5 + 4,5 + 9) of the anatomical distribution of the mucosal damage by showing the differences in terms of architectural damage (**B**), epithelial abnormalities (**C**) and lamina propria cellularity (**D**) between the proximal and the distal part of the colonic mucosa.

**Figure 4 ijms-22-02028-f004:**
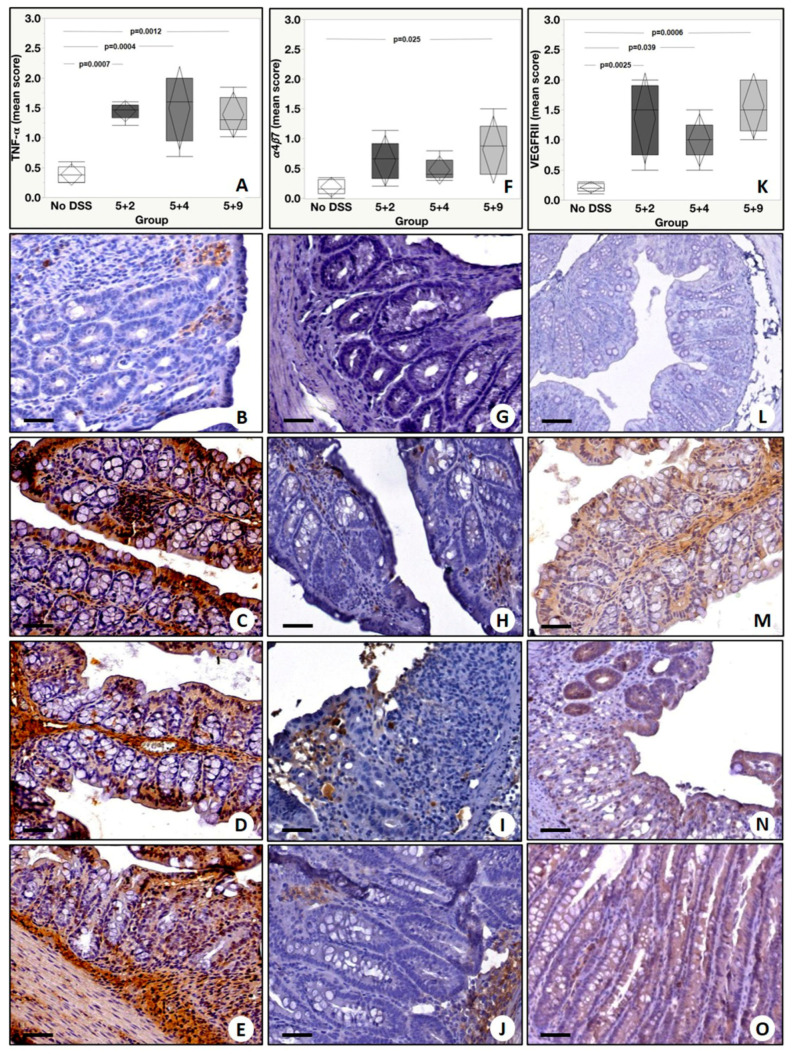
Immunohistochemical evaluation of the expression of TNF-α, α_4_β_7_ and VEGFRII. (**A**) Graph shows the expression of TNF-α among the experimental groups. (**B**) No/rare TNF-α positive inflammatory cells are displayed. (**C**–**E**) Images show several TNF-α positive inflammatory cells in colons from 5 + 2, 5 + 4 and 5 + 9 mice. (**F**) Graph shows the expression of α_4_β_7_ among the experimental groups. (**G**) No/rare α_4_β_7_ positive inflammatory cells are displayed. (**H**) Moderate α_4_β_7_ positive inflammatory infiltrate in a colon from 5 + 2 mouse. (**I**) Rare α_4_β_7_ positive cells in a colon from 5 + 4 mouse. (**J**) High/moderate α_4_β_7_ positive inflammatory cells in a colon from 5 + 9 mouse. (**K**) Graph displays the expression of VEGFRII among the experimental groups. (**L**) Colon of NO DSS mouse shows no/rare VEGFRII positive cells. (**M**) High VEGFRII positive inflammatory infiltrate in a colon from 5 + 2 mouse. (**N**) Image displays some α_4_β_7_ positive cells in a colon from 5 + 4 mouse. (**O**) Moderate VEGFRII positive inflammatory cells in a colon from of 5 + 9 mouse. Scale bar represents 200 µm in each image.

**Figure 5 ijms-22-02028-f005:**
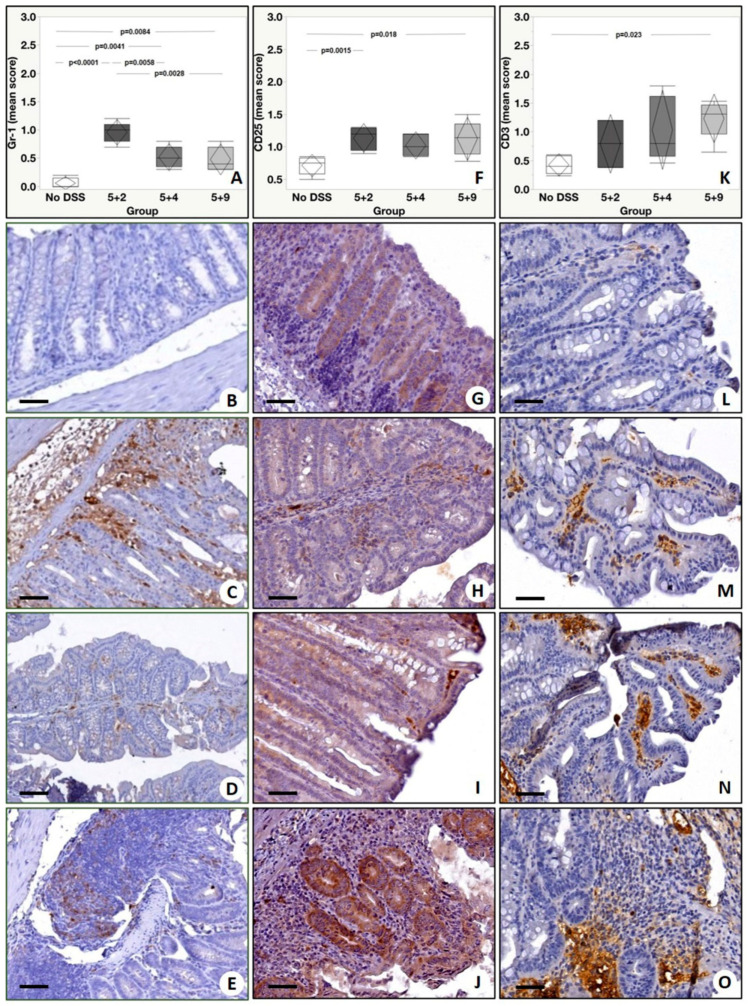
Immunohistochemical evaluation of the expression of Gr-1, CD25 and CD3. (**A**) Graph shows the expression of Gr-1 among the experimental groups. (**B**) No/rare Gr-1 positive inflammatory cells are displayed. (**C**) Numerous Gr-1 positive cells in a colon from 5 + 2 mouse. (**D**,**E**) Images show some Gr-1 positive inflammatory cells in bowels of 5 + 4 and 5 + 9 mice. (**F**) Graph shows the expression of CD25 among the experimental groups. (**G**) Some CD25 positive cells are displayed. (**H**–**J**) High/moderate CD25 positive cells in colons from 5 + 2, 5 + 4 and 5 + 9 mice. (**K**) Graph displays the expression of CD3 among the experimental groups. (**L**) Colon from NO DSS mouse shows no/rare CD3 positive cells. (**M,N**) Numerous CD3 positive cells in colons from 5 + 2 and 5 + 4 mice. (**O**) High CD3 positive cells in a colon from 5 + 9 mouse. Scale bar represents 200 µm in each image.

**Figure 6 ijms-22-02028-f006:**
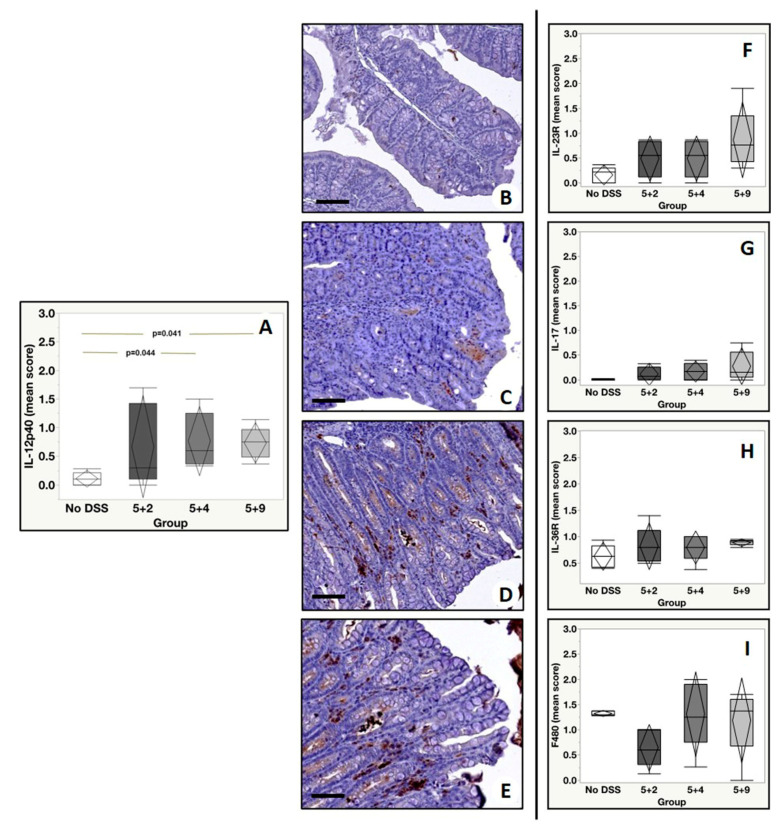
Immunohistochemical evaluation of the expression of IL-12p40, IL-23R, IL-17, IL-36R and F480. (**A**) Graph shows the expression of IL-12p40 among the experimental groups. (**B**) No/rare IL-12p40 positive cells are displayed. (**C**) Some IL-12p40 positive cells in a colon from 5 + 2 mouse. (**D**,**E**) Images show several IL-12p40 positive cells in bowels of 5 + 4 and 5 + 9 mice, respectively. (**F**–**I**) Graph show the expression of IL-23R, IL-17, IL-36R and F480 among the experimental groups. Scale bar represents 200 µm in each image.

**Table 1 ijms-22-02028-t001:** Proposed histological score based on histopathology of human inflammatory bowel disease (IBD).

Score	Lamina Propria Cellularity	Architectural Damage	Epithelial Abnormalities
0	Normal presence of mononuclear inflammatory cells (lymphocytes, plasma cells, eosinophils) in lamina propria.	Normal crypts (straight, parallel and extend from immediately above the muscularis mucosae)	No signs of epithelial damage.
1	Mild increase in mononuclear inflammatory cells in the lamina propria.	Mild crypt distortion (includes loss of parallelism between crypts, variation in size and shape)	Mucin depletion (reduction in number of goblet cells and/or intracellular mucin) and/or repairing/reactive changes (may include nuclear enlargement, loss of nuclear polarity, prominent nucleoli, presence of mitotic figures).
2	Striking increase in mononuclear inflammatory cells and possible presence of a minor neutrophilic component in the lamina propria	Moderate crypt distortion (more evident crypt distortion) and shortening: ≤1/3 of crypt height from muscularis mucosae	Mucin depletion, repairing/reactive changes and erosions (epithelial necrosis involving epithelium mainly superficially).
3	Striking increase in mononuclear inflammatory cells and easily recognizable presence of neutrophils in the lamina propria.	Severe crypt distortion and shortening of ≥1/2 of crypt height from muscularis mucosae (atrophy). Additional evidence: wider spacing of crypts	Mucin depletion, repairing/reactive changes and ulcers (epithelial necrosis involving the total thickness of the mucosa).

**Table 2 ijms-22-02028-t002:** List of antibodies used in the study.

Antibody	Characteristics	Dilution	Retrieval
**Anti-TNF-α**	Rat monoclonal, clone XT3.11; BioXCell, NH, USA	1:1000	Citrate pH 6.0
**Anti-α_4_β_7_**	Rat monoclonal, clone DATK32; BioXCell, NH, USA)	1:700	EDTA–Citrate Ph 7.8
**Anti-IL-12B p40**	Rat monoclonal, clone C17.8; Santa Cruz Biotechnology, Texas, USA	1:100	Citrate pH 6.0
**anti-CD3**	mouse monoclonal clone V9; Ventana, Tucson, AZ, USA	Pre-diluted	EDTA citrate pH 7.8
**anti-CD25**	Rabbit monoclonal, clone SP176; Spring Bioscience, Pleasanton, CA, USA	1:50	EDTA–Citrate pH 7.8
**anti-IL-17A**	Rat monoclonal, clone TC11-18H10; Novus Biologicals, Littleton, CO, USA	1:50	EDTA–Citrate pH 7.8
**anti-Ly6G (Gr-1)**	Mouse monoclonal, clone RB6-85C; Novus Biologicals, Littleton, CO, USA	1:100	EDTA–Citrate pH 7.8
**anti-IL-36R**	IgG2a anti-mouse IL36 with mutated Fc portion Boehringer Ingelheim, Germany	1:100	Citrate pH 6.0
**anti-FLK1 (VEGFRII)**	Rabbit polyclonal; Spring Bioscience, Pleasanton, CA, USA.	1:100	Citrate pH 6.0
**anti-F4/80**	Rat monoclonal, clone CI:A3-1; BioXCell, NH, USA	1:500	EDTA–Citrate pH 7.8
**Anti-IL-23R**	Rabbit polyclonal, Novus Biologicals, Littleton, CO, USA	1:100	Citrate pH 6

## Data Availability

Data will be available on request.

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
