# Peer review of "Extensive Histopathological Characterization of Inflamed Bowel in the Dextran Sulfate Sodium Mouse Model with Emphasis on Clinically Relevant Biomarkers and Targets for Drug Development"

_ijms, 2021, doi:10.3390/ijms22042028_

Round 1
Reviewer 1 Report
The manuscript “Extensive histopathological characterization of inflamed bowel in the DSS mouse model with emphasis on clinically relevant biomarkers and targets for drug development” indicates that the “Knowledge of the spatial-temporal pattern distribution of the pathological lesions of a well-characterized disease model lays the foundation for the study of the tissue expression of meaningful predictive biomarkers, thereby improving translational success rates of preclinical studies for a personalized management of IBD patients”. On the other hand the “data reported allowed to develop an original scoring method useful as a tool for the histological assessment of preclinical models of DSS-induced IBD”.
It is an interesting work that can serve as a reference for future research. As strengths I point out the manuscript as a whole. In fact, is very well written, its sections are presented in a balanced, coherent and in an assertive way. The results are very well presented and supported in figures, including photomicrographs, very informative for area readers. I really liked the discussion because it is based on the available scientific data and that the authors compare with their results, but mainly because it is presented in a very pedagogical and didactic way - which is important, considering the objectives of the work.
Regarding the nuclear approach, and from a conceptual point of view, I do not highlight any weaknesses. However, the quality of photomicrographs should be improved as some aspects mentioned in the captions are not very noticeable, for which another major magnification and some notations (arrows, etc.) could help. These aspects are even more evident for a certain type of reader who is not very familiar with, for example, immunohistochemical techniques.
In conclusion I am of the opinion this work justifies its publication, and which I recommend, in International Journal of Molecular Sciences. However, there are some minor/moderate aspects that I would like to clarify/suggest with the authors before its publication.
Comment 1 (Title): It does not seem appropriate in the title to use the abbreviation DSS for what I suggest: “Extensive histopathological characterization of inflamed bowel in the Dextran Sulfate Sodium mouse model with emphasis on clinically relevant biomarkers and targets for drug development”.
Comment 2 (Key words, line 65): I suggest adding “Dextran Sulfate Sodium” and eventually “mouse”
INTRODUCTION
Comment 3 (lines 73-75): I think this statement justifies bibliographic references “…but it is worth highlighting that there is considerable genotypic and phenotypic variation even within humans with a similar diagnosis of either Crohn’s disease (CD) or ulcerative colitis (UC)”.
Comment 4 (lines 77): I think that the authors meant “…to compare data using a reliable animal model of colitis…” instead “…to compare data using a reliable animal of colitis…”.
RESULTS
Comment 5 (line 98 or 99): As it is the first time that the authors refer to DSS, I suggest writing Dextran Sulfate Sodium (DSS).
Comment 6 (line 108): As it is the first time that the authors refer to DAI, I suggest writing Diseases Activity Index (DAI)
Comment 7 (line 110): As it is the first time that the authors refer to CTRL, I suggest writing Control (CTRL). However it is the only time that it appears because the authors always refer to the Control group as No DSS. If it stays in the text add to the abbreviations.
Comment 8 (line 123): I do not think that the subtitle “Mucosal damage” is necessary, which suggests that there are other.
Comment 9 (line 129): I suggest that the authors standardize the terminology and always use the same criteria. In text appears “lamina propria cellularity, architectural damage and epithelial abnormalities”, in Table 1, lamina propria cellularity, mucosal damage and epithelial abnormalities”, in Figure 1, cellularity, architecture, epithelial damage”.
Comment 10 (line 136, Figure 2): The quality of photomicrographs should be improved as some aspects mentioned in the captions are not very noticeable, for which another major magnification and some notations (arrows, etc.) could help.
Comment 11(lines 137 and 141): “plasma cells” instead “plasmacells”.
Comment 12 (lines 155-166): Please see Comment 9.
Comment 13 (line 227, Figure 3): “…lamina propria cellularity, architectural damage and epithelial abnormalities”. Please see Comment 9.
Comment 14 (line 246, Figure 4): The quality of photomicrographs should be improved as some aspects mentioned in the captions are not very noticeable, for which another major magnification and some notations (arrows, etc.) could help. On the other, and if is the case, I suppose so that should be mentioned “hematoxylin counterstain”.
Comment 15 (line 272, Figure 5): The quality of photomicrographs should be improved as some aspects mentioned in the captions are not very noticeable, for which another major magnification and some notations (arrows, etc.) could help. On the other, and if is the case, I suppose so that should be mentioned “hematoxylin counterstain”.
DISCUSSION
Comment 16 (line 347): Please explain RCU and add to Abbreviations.
Comment 17 (line 420): “…tomography (PET)/Single-photon…”
MATERIALS AND METHODS
Comment 18 (line 447): Please explain “…in collective cages at 22+/21 °C…” or is 22 ± 2ºC ?
Comment 19 (line 455): Please correct the sentence “Control mice (n=5) received normal drinking water throughout”.
Comment 20 (line 455): “Control mice (n=5)…”, but in line 226 (Figure 3) is referred “…control group (No DSS; n= 10 mice/group)”. Were 5 or 10 animals?
Comment 21 (line 457): the authors refer “We performed these experiments several times in different setups (i.e.: by comparing the administration of 3% DSS in male vs. female C57BL/6 mice and by comparing the administration of 2% DSS vs. 3% DSS in female mice; data not shown)”, but do not explain the reason for the choice. At 3% did any animals die or show changes that are too serious and/or potentially lethal? If the authors say that the 2% level is better, it should be said why.
Comment 22 (line 491): It would not have been interesting to have used PAS/Alcian blue? In Table 1 the authors refer Mucin depletion (reduction in number of goblet cells and/or intracellular mucin), with this technique would have been more objective.
Comment 23 (line 496): How were the images obtained? Werte evaluated by direct observation (with what magnification) or in photomicrographs?
Comment 24 (line 505): Table 1
Comment 25 (line 508): One slide/animal?
Comment 26 (line 513): What negative and positive controls are used?
Comment 27 (line 520): Did the authors counter-stained with Hematoxylin? (Please see comments 14 and 15)
Comment 28 (lines 521-522): If in the Results the authors speak of an evident decrease in the number of cells, they must explain the criteria for each of the levels 0, 1, 2, 3 for example: 0, no staining; 1, definite staining of ≤ 25% of cells; 2, definite staining of 26-74% of cells, 3 definite staining of ≥ 75% of cells). Of course, is just an example.
Comment 29 (line 558): I suggest presenting the abbreviations in alphabetical order and checking if any are missing.
Author Response
Manuscript ID: ijms-1075520
“Extensive histopathological characterization of inflamed bowel in the DSS mouse model with emphasis on clinically relevant biomarkers and targets for drug development”.
Submitted:
"International Journal of Molecular Sciences"
REPLY TO REVIEWER 1:
The manuscript “Extensive histopathological characterization of inflamed bowel in the DSS mouse model with emphasis on clinically relevant biomarkers and targets for drug development” indicates that the “Knowledge of the spatial-temporal pattern distribution of the pathological lesions of a well-characterized disease model lays the foundation for the study of the tissue expression of meaningful predictive biomarkers, thereby improving translational success rates of preclinical studies for a personalized management of IBD patients”. On the other hand, the “data reported allowed to develop an original scoring method useful as a tool for the histological assessment of preclinical models of DSS-induced IBD”.
It is an interesting work that can serve as a reference for future research. As strengths I point out the manuscript as a whole. In fact, is very well written, its sections are presented in a balanced, coherent and in an assertive way. The results are very well presented and supported in figures, including photomicrographs, very informative for area readers. I really liked the discussion because it is based on the available scientific data and that the authors compare with their results, but mainly because it is presented in a very pedagogical and didactic way - which is important, considering the objectives of the work.
Regarding the nuclear approach, and from a conceptual point of view, I do not highlight any weaknesses. However, the quality of photomicrographs should be improved as some aspects mentioned in the captions are not very noticeable, for which another major magnification and some notations (arrows, etc.) could help. These aspects are even more evident for a certain type of reader who is not very familiar with, for example, immunohistochemical techniques.
In conclusion I am of the opinion this work justifies its publication, and which I recommend, in International Journal of Molecular Sciences. However, there are some minor/moderate aspects that I would like to clarify/suggest with the authors before its publication.
Reply: We thank the Reviewer for expressing interest and appreciation in our work. As suggested, we have corrected and clarified several sentences and submitted new pictures.
Comment 1 (Title): It does not seem appropriate in the title to use the abbreviation DSS for what I suggest: “Extensive histopathological characterization of inflamed bowel in the Dextran Sulfate Sodium mouse model with emphasis on clinically relevant biomarkers and targets for drug development”.
Reply: Title has now been corrected.
Comment 2 (Key words, line 65): I suggest adding “Dextran Sulfate Sodium” and eventually “mouse”
Reply: We changed the key word as suggested.
INTRODUCTION
Comment 3 (lines 73-75): I think this statement justifies bibliographic references “…but it is worth highlighting that there is considerable genotypic and phenotypic variation even within humans with a similar diagnosis of either Crohn’s disease (CD) or ulcerative colitis (UC)”.
Reply: We added another bibliographic reference that justifies this statement.
Comment 4 (lines 77): I think that the authors meant “…to compare data using a reliable animal model of colitis…” instead “…to compare data using a reliable animal of colitis…”.
Reply: We corrected the mistake.
RESULTS
Comment 5 (line 98 or 99): As it is the first time that the authors refer to DSS, I suggest writing Dextran Sulfate Sodium (DSS).
Reply: Done.
Comment 6 (line 108): As it is the first time that the authors refer to DAI, I suggest writing Diseases Activity Index (DAI)
Reply: Done.
Comment 7 (line 110): As it is the first time that the authors refer to CTRL, I suggest writing Control (CTRL). However, it is the only time that it appears because the authors always refer to the Control group as No DSS. If it stays in the text add to the abbreviations.
Reply: We refer to the Control Group as No DSS also in this line of the text to make it uniform in the entire manuscript.
Comment 8 (line 123): I do not think that the subtitle “Mucosal damage” is necessary, which suggests that there are other.
Reply: With the aim to assess whether our model could be candidate as a reliable, truly translatable, preclinical model of human disease, we performed a spatio-temporal analysis of the established DSS-induced model based on the last Guidelines for the Histopathology of Human IBD. Accordingly, we attempted to perform an accurate differential diagnosis between the two major phenotypic forms of IBD (Ulcerative Colitis and Chron’s Disease) in our model. As reported in the Scoring System Paragraph (Materials and Methods, lines 505-508) we analyzed all mouse colons by light microscopy paying attention to two specific and fundamental pathological aspects: the mucosal damage (i.e. the lesion type) and the anatomical distribution of the lesions. In the paragraph of Results the subtitles “Mucosal damage” and “Anatomical distribution of the lesions” were conceived to guide the reader to appreciate the histopathological analysis performed according to the main diagnostic criteria employed for human IBD. Maybe, it would be clearer to title as follows the subtitles: Mucosal damage and Anatomical distribution of the mucosal damage. We, therefore, changed accordingly.
Comment 9 (line 129): I suggest that the authors standardize the terminology and always use the same criteria. In text appears “lamina propria cellularity, architectural damage and epithelial abnormalities”, in Table 1, lamina propria cellularity, mucosal damage and epithelial abnormalities”, in Figure 1, cellularity, architecture, epithelial damage”.
Reply: We now standardized text, tables and figures with the same terms along the whole manuscript as follows: lamina propria cellularity, architectural damage, epithelial abnormalities.
Comment 10 (line 136, Figure 2): The quality of photomicrographs should be improved as some aspects mentioned in the captions are not very noticeable, for which another major magnification and some notations (arrows, etc.) could help.
Reply: To improve the readability of Fig 2, we added arrows and modified the notations to every single picture in order to underline some of the most relevant histopathological features, described according to Table 1. Moreover, thanks to reviewer’s suggestion we could also note that by inverting Figures D with L reader is better guided in appreciating score criteria with respect the histological features to be considered.
Comment 11(lines 137 and 141): “plasma cells” instead “plasmacells”.
Reply: Done.
Comment 12 (lines 155-166): Please see Comment 9.
Reply: see Reply to comment 9.
Comment 13 (line 227, Figure 3): “…lamina propria cellularity, architectural damage and epithelial abnormalities”. Please see Comment 9.
Reply: see Reply to comment 9.
Comment 14 (line 246, Figure 4): The quality of photomicrographs should be improved as some aspects mentioned in the captions are not very noticeable, for which another major magnification and some notations (arrows, etc.) could help. On the other, and if is the case, I suppose so that should be mentioned “hematoxylin counterstain”.
Reply: Done.
Comment 15 (line 272, Figure 5): The quality of photomicrographs should be improved as some aspects mentioned in the captions are not very noticeable, for which another major magnification and some notations (arrows, etc.) could help. On the other, and if is the case, I suppose so that should be mentioned “hematoxylin counterstain”.
Reply: Done.
DISCUSSION
Comment 16 (line 347): Please explain RCU and add to Abbreviations.
Reply: We apologize for the mistake. We changed with UC (ulcerative colitis).
Comment 17 (line 420): “…tomography (PET)/Single-photon…”
Reply: Done.
MATERIALS AND METHODS
Comment 18 (line 447): Please explain “…in collective cages at 22+/21 °C…” or is 22 ± 2ºC?
Reply: It is 20 ± 2 °C. We corrected it.
Comment 19 (line 455): Please correct the sentence “Control mice (n=5) received normal drinking water throughout”.
Reply: We corrected as follows: “Control group (NO DSS; n=5 mice/group) received normal drinking water ad libitum.”
Comment 20 (line 455): “Control mice (n=5)…”, but in line 226 (Figure 3) is referred “…control group (No DSS; n= 10 mice/group)”. Were 5 or 10 animals?
Reply: As reported in Materials and Method, the experimental protocol (see 462- 463 lines) was performed twice by using, for each experimental, independent run, 5 animals/experimental group (see lines 458-462). Based on clinical results obtained (lines 105-121, Fig 1), we then analysed colons from the 2 experimental runs by pooling data (as reported in line 226, Fig 3).
Comment 21 (line 457): The authors refer “We performed these experiments several times in different setups (i.e.: by comparing the administration of 3% DSS in male vs. female C57BL/6 mice and by comparing the administration of 2% DSS vs. 3% DSS in female mice; data not shown)”, but do not explain the reason for the choice. At 3% did any animals die or show changes that are too serious and/or potentially lethal? If the authors say that the 2% level is better, it should be said why.
Reply: Literary data alert that the model of DSS-induced colitis requires to be set up every time when there is an interest in developing it. Indeed, mice show different susceptibilities and responsiveness to DSS according to a variety of experimental conditions: DSS concentration, DSS molecular weight, duration of DSS exposure, manufacturer, batch, mouse strain, mouse gender, microbiological state, intestinal flora, average room temperature, food, stress, etc. From this standpoint we approached our work starting from the selection of the most suitable DSS-model to ensure consistency and reproducibility to our studies. Among the different setups (i.e.: by comparing the administration of 3% DSS in male vs. female C57BL/6 mice and by comparing the administration of 2% DSS vs. 3% DSS in female mice; data not shown), “we found that the optimal DSS dosage and duration to obtain a slow and steady onset of the disease corresponded to the administration of DSS at 2% for 5 days followed by water ad libitum. “We have also found that DSS-induced disease development occurs more gradually in female animals when compared to male mice, therefore we found easier the management of female mice, since they better tolerate the treatment” (lines 105-110). Indeed, male mice showed a higher incidence of death (also due to a more aggressive behavior in response to DSS-induced stress) and a higher severity of disease (severe rectal bleeding and weight loss within 3-4 days of DSS administration, visible signs of illness, including a hunched back, raised fur, symptoms of sepsis and reduced mobility because of diarrhea and anemia) compared to female mice. In the light of these results, the protocol that we adopted let us to manage a successful and easily reproducible model of DSS-induced colitis for our studies. According to these considerations, we changed the beginning of the text of the subparagraph “Induction and Evaluation of DSS Colitis” (Materials and Methods).
Comment 22 (line 491): It would not have been interesting to have used PAS/Alcian blue? In Table 1 the authors refer Mucin depletion (reduction in number of goblet cells and/or intracellular mucin), with this technique would have been more objective.
Reply: Yes, indeed we evaluated mucin depletion based on morphological evaluation made by two expert pathologists blindly. It is true: PAS/Alcian blue could help in evaluating variation in mucin cellular content. Nevertheless, normal intestinal cells are characterized by a polar organization (nucleus laying on basal lamina and mucinous cytoplasm pointing towards the glandular lumen). An impaired mucinous compartment respect to the nucleus (judged in an IBD context) is morphologically consistent with mucus depletion. Thus, morphological features are, in this context, sufficient to evaluate with scientific rigor hematoxylin and eosin slides (easier to manage).
Comment 23 (line 496): How were the images obtained? Were evaluated by direct observation (with what magnification) or in photomicrographs?
Reply: We digitalized all slides (Iscan Coreo, Ventana, Tucson, AZ, USA) and images were captured using ImageView Software. Immunohistochemical positivity was evaluated on digital images at 20x magnification. We added this information in Materials and Methods.
Comment 24 (line 505): Table 1
Reply: Done
Comment 25 (line 508): One slide/animal?
Reply: The entire colon of each animal was subdivided into rings of 0.5 cm (line 490) which were analyzed on three levels (lines 504-505). See also in the Discussion “in agreement with European Crohn's and Colitis Organisation (ECCO) guidelines we performed multiple colon biopsies (in our case we sampled the entire colon) keeping track of the anatomical site of origin of each biopsy (thanks to the employment of a handicraft tissue array) and by examining multiple sections (n=3) for each biopsy since lesions may be focal”. Depending on the colon length we analyzed 10-16 rings/each colon (about 30-48 slides/animal).
Comment 26 (line 513): What negative and positive controls are used?
Reply: To assess the background of immuno-staining, for each reaction we included a negative control by incubating the sections with secondary antibodies (HRP) and detection system (DAB). According to the manufacturer’s instructions, reactions have been set-up by using specific control tissues. We added this information at the end of the sub-paragraph “Immunohistochemistry” (Materials and Methods).
Comment 27 (line 520): Did the authors counter-stained with Hematoxylin? (Please see comments 14 and 15)
Reply: We added this missing information in the subparagraph about immunohistochemistry (Materials and Methods, line 524).
Comment 28 (lines 521-522): If in the Results the authors speak of an evident decrease in the number of cells, they must explain the criteria for each of the levels.
Reply: We modified the text (line 526-533) as follows “Immunohistochemical positivity was evaluated on digital images by two blind observers who assigned a score (0-3) by counting the number of positive cells in 4 fields at 20x (score 0: positive cells ≤2; score 1: 3< positive cells ≤30; score 2 (31 < positive cells ≤100), score 3 (positive cells > 100). Considering the intensity and the number of positive cells detected following the immunoreactions for TNF-α e CD3 the score (0-3) was attributed by establishing different cut-offs (score 0: positive cells ≤2; score 1: 3< positive cells ≤50; score 2 (51 < positive cells ≤200), score 3 (positive cells > 200)”.
Comment 29 (line 558): I suggest presenting the abbreviations in alphabetical order and checking if any are missing.
Reply: We ordered the abbreviation list as suggested.
Reviewer 2 Report
The manuscript is interesting and well written. However, there are several concerns, particularly with regard to the experimental design.
1) How were the "clinically relevant" biomarkers determined? There were no biomarkers for Th1, Th2 cells or T cells that may have undergone plasticity (e.g., GMCSF). There were no intracellular targets (e.g., DHODH, Ror-gamma T). The criteria for choosing the markers needs to be discussed more in the paper.
2) A five day (DSS only) group needs to be included in the experimental design. This provides the real baseline for the three DSS plus water treatment groups. Data from this time point would also provide valuable/relevant biomarker data.
3) The IL-12/IL-23 axis is a clear clinical target for IBD patients. Drugs like Ustekinumab are already approved. Prominent changes were not seen in markers of this axis. Explanation and discussion is not included in the manuscript.
4) In order to validate this novel DSS colitis model, some drug testing is clearly needed. In this regard, corticosteroids are not effective in the acute DSS model, but are effective in chronic DSS models involving multiple cycles of DSS. Since the timing of DSS exposure is a bit different with this model, testing with a standard IBD drug is needed for model validation.
5) The statement in line 99 about this being the most reliable DSS model is too strong. This comment relates to my critiques in responses 2,3, and 4 above.
Author Response
Manuscript ID: ijms-1075520
“Extensive histopathological characterization of inflamed bowel in the DSS mouse model with emphasis on clinically relevant biomarkers and targets for drug development”.
Submitted:
"International Journal of Molecular Sciences"
The manuscript is interesting and well written. However, there are several concerns, particularly with regard to the experimental design.
Reply: We would like to thank the Reviewer 2 for expressing interest in our work and for her/his useful suggestions.
1) How were the "clinically relevant" biomarkers determined? There were no biomarkers for Th1, Th2 cells or T cells that may have undergone plasticity (e.g., GMCSF). There were no intracellular targets (e.g., DHODH, Ror-gamma T). The criteria for choosing the markers needs to be discussed more in the paper.
Reply: We thank the reviewer for this comment that helps us to better explain the rationale of our study. According to what reported in the Introduction, the secondary aim of the study was oriented to use the DSS model in order to characterize the in-situ expression of clinically relevant biomarkers and targets involved in IBD. Recent advances in imaging technology continue to improve the ability of imaging techniques to non-invasively monitor disease activity and treatment response in preclinical models and in humans with IBD. Nevertheless, to date very few imaging modalities are routinely available (i.e. CT, NMR, radiolabelled-WBC, FDG PET/CT). It is, therefore, highly clinically relevant to identify new biomarkers that can be target for new radiopharmaceuticals. Amongst the many targets we could possibly investigated, we analysed, in DSS model, some of those already well known and involved in pathogenesis or therapy of human IBD, against which we can produce a radiopharmaceutical. This is why we excluded intracellular targets, or most of soluble targets.
Accordingly, the selection of “clinically relevant biomarkers” has now been clarified in the text (see Introduction: lines 92-98 and Discussion: lines 389-391 and 440-445).
2) A five-day (DSS only) group needs to be included in the experimental design. This provides the real baseline for the three DSS plus water treatment groups. Data from this time point would also provide valuable/relevant biomarkers.
Reply: Before setting this study, we run a few pilot experiments and noticed that animals need a couple of days rest after DSS to be used for our studies and mucosal damage after 5 days therapy is less pronounced that after 2 days recovery. Finally, from data available in literature, it appears that few days of rest after DSS is standard practice. These are the reasons for choosing our protocol.
3) The IL-12/IL-23 axis is a clear clinical target for IBD patients. Drugs like Ustekinumab are already approved. Prominent changes were not seen in markers of this axis. Explanation and discussion is not included in the manuscript.
Reply: Indeed, the choice of these markers is justified by their clinical relevance and we also were surprised that no difference was observed between control mice and DSS mice. Therefore, and thanks to your comment, we asked our statistician to double check all data and analysis and he found an error, for which we apologize. There is, indeed, a statistically significant difference in IL12p40 expression between control mice and DSS mice 5+4 (p=0.044) and between control mice and DSS mice 5+9 (p=0.041) as it clearly appears from figure 6. Nevertheless, as far as IL23R expression is concerned, despite a trend of increase in DSS mice, we found no statistically significant differences. This finding is not surprising. Please consider that Ustekinumab is a mAb against IL-12/IL-23 p40 protein and not against the cytokine receptor. Therefore, the finding of an increase of IL12p40 but not of IL23R is in agreement with what observed in human IBD.
The paper and figure 6, have been corrected accordingly.
4) In order to validate this novel DSS colitis model, some drug testing is clearly needed. In this regard, corticosteroids are not effective in the acute DSS model, but are effective in chronic DSS models involving multiple cycles of DSS. Since the timing of DSS exposure is a bit different with this model, testing with a standard IBD drug is needed for model validation.
Reply: Thank you for the suggestion. It is certainly interesting but it was out of our scope since we focused our attention on the characterization of an acute DSS-induced colitis. We are now planning interventional studies.
5) The statement in line 99 about this being the most reliable DSS model is too strong. This comment relates my critiques in responses 2, 3 and 4 above.
Reply: We modified the beginning of the text in Results to better clarify, as suggested.